# Evaluation of CHROMagar™ LIN-R for the Screening of Linezolid Resistant Staphylococci from Positive Blood Cultures and Nasal Swab Screening Samples

**DOI:** 10.3390/antibiotics11030313

**Published:** 2022-02-25

**Authors:** Delphine Girlich, Liliana Mihaila, Vincent Cattoir, Frédéric Laurent, Christine Begasse, Florence David, Carole-Ann Metro, Laurent Dortet

**Affiliations:** 1INSERM UMR1184-Team RESIST, Faculty of Medicine, Paris-Saclay University, 94270 Le Kremlin-Bicêtre, France; delphine.girlich@universite-paris-saclay.fr; 2Bacteriology-Hygiene Unit, Assistance Publique-Hôpitaux de Paris, Bicêtre Hospital, 94270 Le Kremlin-Bicêtre, France; liliana.mihaila@aphp.fr (L.M.); christine.begasse@aphp.fr (C.B.); florence.david@aphp.fr (F.D.); carole-ann.metro@aphp.fr (C.-A.M.); 3Department of Clinical Microbiology and National Reference Center for Enterococci, University Hospital of Rennes, 35033 Rennes, France; vincent.cattoir@chu-rennes.fr; 4National Reference Center for Staphylococci, Hospices Civils de Lyon, 69002 Lyon, France; frederic.laurent@univ-lyon1.fr; 5Associated French National Reference Center for Antibiotic Resistance: Carbapenemase-Producing Enterobacteriaceae, 94270 Le Kremlin-Bicêtre, France

**Keywords:** screening, oxazolidinones, *Staphylococcus*, prevalence

## Abstract

The increasing number of nosocomial pathogens with resistances towards last resort antibiotics, like linezolid for gram positive bacteria, leads to a pressing need for screening and, consequently, suitable screening media. Some national guidelines on infection prevention (e.g., in Germany) have already recommended screening for linezolid-resistant bacteria, despite an accurate screening medium that was not available yet. In this study, we analyzed the performance and reliability of the first commercial chromogenic medium, CHOMagar™ LIN-R, for screening of linezolid-resistant gram-positive isolates. Thirty-four pure bacterial cultures, 18 positive blood cultures, and 358 nasal swab screening samples were tested. This medium efficiently detected linezolid-resistant *S. epidermidis* isolates from pure bacterial cultures and from positive blood cultures with a high sensitivity (100%) and specificity (100%). Among the 358 nasal swab screening samples prospectively tested, 10.9% were cultured with linezolid-resistant isolates (mostly *S. epidermidis*). Of note, slight growth was observed for 7.5% samples with linezolid-susceptible isolates of *S. epidermidis* (*n* = 1), *S. aureus* (*n* = 1), *Enterococcus faecalis* (*n* = 4), *Lactobacillus* spp. (*n* = 3), gram negatives (*n* = 18). Moreover, few *Candida* spp. also cultured on this medium (1.4%).

## 1. Introduction

Linezolid (LZD) was the first molecule of the oxazolidinone family that was approved by the US Food and Drug Administration for commercial use in 2000. It corresponds to one of the last resort antibiotics to treat infections caused by methicillin-resistant *Staphylococcus aureus* (MRSA) and coagulase-negative staphylococci, vancomycin-resistant enterococci (VRE), and a few other resistant gram-positive bacteria. Oxazolidinones inhibit bacterial growth by interfering with protein synthesis through binding to the 23S rRNA. Consequently, linezolid has been demonstrated to possess antibacterial activity both in vivo and in vitro [1] Thus, linezolid treatment is indicated for the treatment of nosocomial pneumonia and skin and soft tissue infections caused by some gram-positive bacteria [2,3].

In 2009, only 0.34% resistance to linezolid was reported on 6414 isolates of *S. aureus*, coagulase-negative staphylococci (CoNS), *Enterococcus* spp., *Streptococcus pneumoniae*, *viridans* group streptococci, and β-hemolytic streptococci from 56 medical centers in the United States [3]. Until 2007, only sporadically and infrequently have linezolid-resistant (LZD^R^) enterococci been identified by the German National Reference Center for Staphylococci and Enterococci. Nevertheless, over the past three years, a dramatic increase of LZD resistance has been observed in E. faecium and S. epidermidis isolates in Germany and neighboring countries, while the number of LZD^R^ S. aureus remained stable [4]. In both staphylococci and enterococci, resistance to linezolid corresponds to mutations in several alleles of the 23S rRNA encoding gene, mutations in ribosomal proteins L3, L4, and L22 [5], and to the acquisition of plasmids carrying linezolid resistance genes such as *cfr* [6], *optrA* [7] or *poxtA* [8]. The *cfr* gene encodes a 23S rRNA methylase that confers resistance to linezolid, phenicols, lincosamides, pleuromutilins, and spectrogramins, but not to tedizolid [9]. It has been reported in staphylococci, enterococci, *Streptococcus suis*, *Bacillus* spp., but also in several gram-negative bacteria such as *Escherichia coli* [10]. More recently, the *poxtA* gene, encoding an ATP-binding cassette (ABC) protein, has been detected in a clinical methicillin resistant *Staphylococcus aureus* (MRSA) in Italy [11] and seems to be already widely prevalent among enterococci in Portugal, Italy, Denmark, China [8], Ireland [12], and Tunisia [7]. Usually, *poxtA* is part of a composite transposon-like structure containing IS1216 elements [8] that has been demonstrated to have the qualities of self-excision and circularization leading to their dissemination [13]. Regarding the detection of the most worrisome LZD resistance mechanisms (the plasmid-encoded determinant), Bender et al. developed a multiplex-PCR able to simultaneously detect *cfr, optrA* or *poxtA* genes [14]. However, this kind molecular method cannot be performed on all isolates routinely. Only LZD^R^ staphylococci and LZD^R^ enterococci from clinical or screening samples should be tested. Accordingly, it is crucial to develop selective screening media to prevent dissemination of LZD^R^ staphylococci and enterococci. Currently, the media developed for the screening of LZD^R^ gram positives were Enterococcoselagar™ supplemented with linezolid at 2 mg/L [15], or a home-made medium supplemented with linezolid (1.5 mg/L) and anti-gram negatives and yeast (aztreonam, colistin and amphothericin B) (SuperLinezolid medium) [16]. Both of them demonstrated high sensitivity >96.6% and specificity >94.4%. However, the SuperLinezolid medium was mainly tested on bacterial cultures of LZD^R^ *S. epidermidis* with a sensitivity of 82% at 24 h that reached 100% at 48 h. Additionally, this medium also demonstrated a quite low detection limit for LZD^R^ enterococci. LZD supplemented Enterococcoselagar^TM^ was only tested for the screening of enterococci from rectal swabs or stool samples. The authors demonstrated that 48 h incubation is required for a reliable detection of LZD^R^ enterococci.

CHROMagar™ LIN-R (CHROMagar, Paris, France) is a selective, commercially available medium designed and focused on linezolid-resistant bacteria. It is suitable for identifying LZD^R^ gram positives from clinical samples such as nasal swabs used for MRSA screening and rectal swabs performed for the screening of glycopeptide resistant enterococci. This medium was designed to inhibit the growth of yeast, gram negatives and linezolid susceptible isolates. Chromogenic molecules were included to discriminate *Staphylococcus* spp. (pink) and *Enterococcus* spp. (blue). CHROMagar™ LIN-R has been recently validated by Layer et al. on a collection of well-characterized isolates of staphylococci and enterococci and showed excellent performances to identify LZD resistance isolates [17], but the performances of this medium were not evaluated on clinical samples. Here, we evaluated the performances of the CHROMagar™ LIN-R medium on characterized isolates of LZD^R^ staphylococci including *cfr* positive strains [18]. This medium was also evaluated prospectively on clinical samples (positive blood cultures and nasal screening samples) recovered from patients hospitalized in wards previously demonstrated to have a high prevalence of LZD^R^ staphylococci [18].

## 2. Results

### 2.1. Performance of CHROMagar™ LIN-R on Pure Isolates

All of the tested LZD^R^ *S. epidermidis* (MICs > 4 µg/mL) were grown on CHROMagar™ LIN-R, resulting in pink colonies that were as large and numerous as those growing on the Mueller-Hinton agar plates (Appendix A, panel A and Table 1). This result was independent of the *cfr* gene content done in a previous work [18]. LZD^R^ *S. aureus* grew on CHROMagar™ LIN-R also resulted in pink colonies, whereas *E. faecium* grew on CHROMagar™ LIN-R giving blue colonies (Appendix A, panel A). None of the twelve LZD^S^ staphylococci from five different species nor the two LZD^S^ *E. faecium* grew on CHROMagar™ LIN-R (Table 1). It was assumed that the traces observed at 48 h of culture with susceptible strains should not be considered as a real growth (Appendix A, Panel B). On pure isolates, this medium had a high sensitivity of 100% (95% CI 67.8–100%) and a high specificity of 100% (95% CI 65.5–100%) (Table 1).

### 2.2. Performance of CHROMagar™ LIN-R on Positive Blood Cultures with Grape Shaped Gram Positives

Among the 18 samples tested, nine corresponded to LZD^R^ *S. epidermidis* isolates that grew as confluent pink colonies of 0.5 mm after 24 h to 1 to 2 mm after 48 h incubation. Six samples gave rise to less than five white/pink colonies on CHROMagar™ LIN-R. These isolates were identified as LZD^S^ *S. epidermidis* (*n* = 4), *S. warneri* (*n* = 1), and *Micrococcus luteus* (*n* = 1). A very slight growth was found on CHROMagar™ LIN-R as compared to MH agar and blood agar. This suggests that the slight growth observed on CHROMagar™ LIN-R was most probably due to an inoculum artifact and should not be considered, as shown with pure cultures (Appendix A). Indeed, the bacterial inoculum in a positive blood culture is usually comprised of between 5.10^8^ and 10^10^ CFU/mL [19]. No other colony grew on CHROMagar™ LIN-R either at 24 h nor after 48 h incubation at 37 °C. To test this hypothesis, LZD^S^ *S. epidermidis, S. capitis* and *S. hominis* were spiked in remnant negative blood cultures samples and incubated at 37 °C in a BactAlert system. A CHROMagar™ LIN-R medium plated with those positive spiked blood cultures showed a thin growth with the LZD^S^ *S. epidermidis* after 24 h of incubation with no supplementary growth after 48 h (Appendix A). No growth was observed with LZD^S^ *S. capitis* and LZD^S^ *S. hominis*. Using serial dilutions of pure culture of LZD^S^ *S. epidermidis,* we determined that this inoculum artifact could be observed as soon as the bacterial concentration reached 10^8^ CFU/mL (Appendix A). This result is in accordance with those of Nordmann et al., demonstrating growth of LZD^S^ *S. epidermidis* on their home-made LZD-containing medium (Superlinezolid) when the inoculum was high [16].

Thus, without considering the inoculum artifacts, the performances of the CHROMagar™ LIN-R were 100% (95% CI 62.9–100%) and had a high specificity of 100% (95% CI 62.9–100%) (Table 1).

The overall performances of this screening medium for studies on pure isolates and blood cultures are a high sensitivity of 100% (95% CI 80–100%) and 100% (95% CI 79.9–100%) of sensitivity and specificity, respectively (Table 1).

### 2.3. Performance of CHROMagar™ LIN-R on Nasal Swab Screening Samples

Linezolid resistance in *S. epidermidis* has been previously reported to be highly prevalent in this hospital, which has been subjected to a long-lasting successful dissemination of LZD^R^ *S. epidermidis* despite stewardship measures, leading to a 4- to 21-fold higher prevalence of LZD^R^ *S. epidermidis* isolation compared to another close hospital [18]. Among the 358 collected samples tested, 10.9% (39/358) were efficiently cultured with LZD^R^ isolates on CHROMagar™ LIN-R. They included 29 LZD^R^ *S. epidermidis* and 10 LZD^R^ *Corynebacterium tuberculosis* (Table 2). To that end, this CHROMagar^TM^ LIN-R medium could help to recover and control the dissemination of those strains. In contrast to the reported “excellent specificity” of the SuperLinezolid medium [16], on which no false-positive isolate was recovered from spiked stool samples, a defect in specificity was observed with the CHROMagar^TM^ LIN-R medium on which 2.5% (9/358) of the samples grew with LZD^S^ gram positive isolates (Table 2). Regarding the detection of LZD^R^ Enterococcus on linezolid complemented EnterococcoselAgar, Werner et al. already demonstrated that 5.9% (20/336) rectal swab samples grew with LZD^S^ Enterococcus spp. (MICs of ≤2 to 4 mg/L). In addition, it is not surprising that 6.4% (23/358) of the samples were cultured with irrelevant gram negatives (*n* = 18) and yeasts (*n* = 5) (Table 2). However, MALDI-TOF identification rapidly eliminated these discordant colonies from further investigations. This kind of irrelevant culture is not uncommon in screening media. Finally, except for four samples, no culture was observed after only 24 h incubation, suggesting the necessity of 48 h incubation to accurately detect LZD^R^ gram positives. It has also been previously reported that 48 h incubation was required for the SuperLinezolid medium [16].

## 3. Discussion

No commercially available medium has been proposed for the screening of LZD^R^ gram positives. Of note, the two previously home-made media that have been proposed for the screening of linezolid resistance have only been evaluated on rectal swabs and stool samples for the screening of LZD^R^ enterococci [15]. Despite the fact that screened LZD^R^ enterococci remain interesting to limit their dissemination, the resistance to oxazolidinone is more worrisome in staphylococci, especially in *S. epidermidis* [20,21,22], with regard to avoiding the spread of such resistance in MRSA for as long as possible. Accordingly, we demonstrated that the CHROMagar™ LIN-R medium might be an accurate screening medium for the early detection of LZD^R^ gram positives recovered from positive blood cultures but also nasal swab screening samples, especially in medical structures where the prevalence of linezolid resistance is suspected to be high. Of note, such screening media usually possess good sensitivity but might have less specificity. Accordingly, bacterial colonies culturing on such media have to be identified and further subjected to complementary tests to move out false positive results. This is also the case with the CHROMagar™ LIN-R medium for which rapid identification would help to immediately detect the 6.4% of the samples that were cultured with irrelevant intrinsically LZD^R^ gram negatives and yeasts. Susceptibility testing also needed to be performed to confirm the linezolid resistance on the cultured gram positives using an accurate method [23].

Of note, a limitation of this study lies in the fact that we could not truly assess the sensitivity of the CHROMagar™ LIN-R medium on nasal swabs since it was not feasible to perform linezolid susceptibility testing on all staphylococcal colonies that grew on non-selective medium in parallel with the CHROMagar™ LIN-R medium (positive or negative in culture). However, this “real-life” study indicated that the ready to use CHROMagar™ LIN-R medium was easy to handle, and positive cultures with chromogenic compounds are easy to include in a routine workflow. Accordingly, this medium might help to detect high levels of LZD resistance leading to proposed infection control measures in order to contain the dissemination of such a resistance trait (e.g., limitation of linezolid prescription to senior physicians). It might also help to assess an up-to-date prevalence of the linezolid resistance not only in bacterial isolate responsible for infection [24,25] but also in the main reservoir of such resistance, the skin microbiota.

Despite the rather low specificity, this culture medium could be a powerful tool for the screening of LZD^R^ isolates in clinical samples. Further investigations such as bacterial identification and determination of MICs of linezolid (E-test) should be implemented on growing colonies.

## 4. Materials and Methods

### 4.1. Bacterial Isolates

The validation of the CHROMagar™ LIN-R was performed on thirty-four pure bacterial strains collected from a previous study with known genetic backgrounds and plasmid content of *cfr* genes [18] and from the French Reference National Centers for antibiotic resistant staphylococci (Lyon, France), and enterococci (Rennes, France), as follows: linezolid-resistant (LZD^R^) isolates of *S. epidermidis* (*n* = 14), *S. aureus* (*n* = 1), *E. faecium* (*n* = 5); linezolid-susceptible (LZD^S^) isolates of *S. epidermidis* (*n* = 3), *S. aureus* (*n* = 5), *S. capitis* (*n* = 2), *S. hominis* (*n* = 1), *S. caprae* (*n* = 1), and *E. faecium* (*n* = 2) and (Table 1).

### 4.2. Clinical Samples

Regarding the validation of the CHROMagar™ LIN-R on positive blood cultures, 18 consecutive blood cultures positive with grape shaped gram-positives recovered from patients hospitalized in the Paul Brousse hospital from 3 December 2019 to 3 January 2020 were tested. Regarding the validation of the CHROMagar™ LIN-R on nasal samples, all consecutive nasal swabs (*n* = 358) collected for the screening of methicillin resistant *Staphylococcus aureus* at the Bicêtre bacteriology-hygiene unit over a period of 2 months from May to June 2020 were tested prospectively.

### 4.3. CHROMagar™ LIN-R Inoculation Protocols

For the validation on pure bacterial colonies, Bacterial cultures were performed in 4 mL of Brain Heart Infusion for 2 h at 37 °C until reaching an OD_600nm_ of 0.1. A 1/10 dilution was performed in sterile water and 10 µL of this solution was plated on CHROMagar™ LIN-R. Regarding the validation on the blood cultures, 2 drops (ca. 80 µL) of positive blood cultures were directly spread on CHROMagar™ LIN-R. Blood agar was also inoculated to further perform antimicrobial susceptibility testing on Mueller-Hinton (MH) agar (Biorad, Marnes-la-Coquette, France). Nasal swabs were directly spread on CHROMagar™ LIN-R. As a growth control, the nasal swabs were also inoculated on MH agar and on ChromID^®^ MRSA (bioMérieux, La Balmes les Grottes, France). Growth, colony size and color were determined after 24 and 48 h at 37 °C.

### 4.4. Bacterial Identification and Susceptibility Testing

All isolates that grew on CHROMagar™ LIN-R were directly identified using MALDI-TOF (Brucker Daltonics, Bremen, Germany). Antimicrobial susceptibility testing was performed by the disc diffusion method and LZD MIC was assessed by Etest (bioMérieux, La Balme-les-Grottes, France). Results were interpreted according to the EUCAST breakpoint as updated in 2020.

### 4.5. Evaluation of the Inoculum Artifact

A few LZD^S^ isolates were found to grow slightly on the CHROMagar™ LIN-R medium after 48 h incubation (Appendix A, panel B). Accordingly, LZD^S^ *S. epidermidis* (MIC 0.75 µg/mL), LZD^S^
*S. capitis* (MIC 1 µg/mL)*,* and LZD^S^ *S. hominis* (MIC 0.75 µg/mL) were spiked in remnant negative blood cultures samples and incubated at 37 °C in a BactAlert system (BioMerieux). Two drops of the positive spiked-blood cultures were plated on CHROMagar™ LIN-R and dilutions were plated on MH agar for colony counting. In order to evaluate the inoculum effect for pure culture, a suspension of colonies of a fresh overnight culture of an isolate of LZD^S^ *S. epidermidis* (MIC 0.75 µg/mL) was adjusted at 0.5 McFarland and 1/10 dilutions were serially prepared in sterile water. From each dilution, 10 µL were plated on CHROMagar™ LIN-R and MH agar and incubated for 24 h at 37 °C before CFU counting (Appendix A).

## Figures and Tables

**Table 1 antibiotics-11-00313-t001:** Performances of CHROMagar™ LIN-R on pure isolates and on blood cultures of staphylococci and *E. faecium* isolates with 24 h and 48 h incubation.

Strain Description	CHROMagar™ LIN-R			Mueller-Hinton		
	Colony Color	Size and Quantity		Colony Color	Size and Quantity	LZD MICs (mg/L)
		24 h	48 h			
**Pure isolates of staphylococci and *E. faecium* (*n* = 34)**						
*S. epidermidis* LZD^R a^, *n* = 14	pink	0.5 mm Q+ ^b^	1 mm Q+	white	Q+	>4
*S. aureus* LZD^R^, *n* = 1	pink	0.5 mm Q+	1 mm Q+	white	Q+	8
*E. faecium* LZD^R^, *n* = 5	blue	0.5 mm Q+	1 mm Q+	white	Q+	12 and >256
*S. epidermidis* LZD^S^, *n* = 3	NA ^c^	0	0	white	Q+	≤1
*S. aureus* LZD^S^, *n* = 5	NA	0	0	white	Q+	≤2
*S. capitis* LZD^S^, *n* = 2	NA	0	0	white	Q+	≤1
*S. hominis* LZD^S^, *n* = 1	NA	0	0	white	Q+	0.75
*S. caprae* LZD^S^, *n* = 1	NA	0	0	white	Q+	1
*E. faecium* LZD^S^, *n* = 2	NA	0	0	white	Q+	≤2
**Positive blood cultures with grape shaped gram positives (*n* = 18)**						
*S. epidermidis* LZD^R^, *n* = 9	pink	0.5 mm Q+	1.5 mm Q+	white	1 mm Q+	
*S. epidermidis* LZD^S^, *n* = 4	white/pink	0.5 mm q ^d^	1 mm q	white	1 mm Q+	
*S. warneri* LZD^S^, *n* = 1	white/pink	0.5 mm q	1 mm q	white	1 mm Q+	
*Micrococcus luteus* LZD^S^, *n* = 1	white	0.5 mm q	1 mm q	yellow	1 mm Q+	
*S. epidermidis* LZD^S^, *n* = 1	NA	0	0	white	1 mm Q+	
*S. haemolyticus* LZD^S^, *n* = 1	NA	0	0	white	1 mm Q+	
*S. hominis* LZD^S^, *n* = 1	NA	0	0	grey	1 mm Q+	
**Sensitivity for pure isolates and blood cultures 100% (95% CI 80–100%)** **Specificity for pure isolates and blood cultures 100% (95% CI 79.9–100%)**						

^a^ *S. epidermidis* LZD^R^ of ST2, ST5, or ST22 harboring the *cfr* gene or not [18]; ^b^ « Q+ » was used for confluent culture, « q » was used for 1 to 5 colonies detected; ^c^ NA, not applicable; ^d^ Growth most probably due to an inoculum artifact.

**Table 2 antibiotics-11-00313-t002:** Isolates cultured on CHROMagar^TM^ LIN-R from the 358 nasal screening samples.

Sample *n*	Identification of Strains/Resistance	Bacterial Colonies on CHROMagar™ LIN-R			LZD MIC (mg/L)	
		Color	Size and Quantity		24 h Incubation	48 h Incubation
			24 h Incubation	48 h Incubation		
**Targeted LZD^R^ gram positives (*n* = 39)**						
(*n* = 28)	*Staphylococcus epidermidis* LZD^R^	pink	-	1.5 mm q to **Q+**	24 to >256	>256
(*n* = 1)	*Staphylococcus epidermidis* LZD^R^	pink	1 mm **Q+**	1.5 mm **Q+**	24	>256
(*n* = 10)	*Corynebacterium tuberculosis* LZD^R^	pink	-	1.5 mm q to **Q+**	>256	>256
**LZD^S^ gram positives (*n* = 9)**						
(*n* = 1)	*Staphylococcus epidermidis* LZD^S^	pink	-	2 mm **Q+**	1	1.5
(*n* = 4)	*Enterococcus faecalis* LZD^S^	blue	-	0.5 mm **Q+**	1.5	2
(*n* = 2)	*Lactobacillus gasseri* LZD^S^	blue	-	0.5 mm **Q+**	1–2	1.5–3
(*n* = 1)	*Lactococcus lactis* LZD^S^	blue	-	0.5–2 mm **Q+**	1.5	2
(*n* = 1)	*Staphylococcus aureus* LZD^S^	yellow	-	1.5 mm **Q+**	1.5	2
**gram negatives (*n* = 18)**						
(*n* = 1)	*Achromobacter xylosoxidans*	white	-	1 mm **Q+**	NA	NA
(*n* = 1)	*Acinetobacter baumannii*	white	-	3 mm q	NA	NA
(*n* = 3)	*Enterobacter cloacae complex*	blue	-	2 mm q	NA	NA
(*n* = 1)	*Enterobacter cloacae complex*	blue	1.5 mm q	2 mm q	NA	NA
(*n* = 1)	*Escherichia coli*	blue	-	2 mm q	NA	NA
(*n* = 1)	*Klebsiella aerogenes*	blue	-	2 mm q	NA	NA
(*n* = 6)	*Klebsiella pneumoniae*	blue	-	2 mm q	NA	NA
(*n* = 1)	*Klebsiella pneumoniae*	blue	1.5 mm q	2 mm q	NA	NA
(*n* = 2)	*Pseudomonas aeruginosa*	pink	-	2 mm q	NA	NA
(*n* = 1)	*Pseudomonas aeruginosa*	pink	1.5 mm q	2 mm q	NA	NA
**Fungi (*n* = 5)**						
(*n* = 4)	*Candida tropicalis*	pink	-	0.5 mm **Q+**	NA	NA
(*n* = 1)	*Candida orthopsilosis*	pink	-	1 mm **Q+**	NA	NA
**Negative culture (*n* = 287)**						
**Prevalence of LZD^R^ gram positives: 10.9% (CI_95%_ 7.9–14.7%)**						
**Specificity of the CHROMagar™ LIN-R: 89.9% (CI_95%_ 86–92.9%)**						

Note « Q+ » was used for a confluent culture, « q » was used for 1 to 5 colonies detected.

## Data Availability

Not applicable.

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
