# Peer review of "Evaluation of CHROMagar™ LIN-R for the Screening of Linezolid Resistant Staphylococci from Positive Blood Cultures and Nasal Swab Screening Samples"

_antibiotics, 2022, doi:10.3390/antibiotics11030313_

Round 1

Reviewer 1 Report

Technically well done.

However, there is no comparison with other species of 

linezolid resistance (e.g. S. epidermidis). Furthermore, the discussion already stresses 

that the authors could not truly assess the sensitivity of the CHROMagarLIN-R medium on nasal swabs since it was not feasible to perform linezolid susceptibility testing on all staphylococcal colonies that grew on non-selective medium in parallel with the CHROMagarLIN-R medium (positive or negative in culture). 

Author Response

Technically well done.

However, there is no comparison with other species of linezolid resistance (e.g. S. epidermidis). Furthermore, the discussion already stresses that the authors could not truly assess the sensitivity of the CHROMagar™ LIN-R medium on nasal swabs since it was not feasible to perform linezolid susceptibility testing on all staphylococcal colonies that grew on non-selective medium in parallel with the CHROMagar™ LIN-R medium (positive or negative in culture).

Answer: We want to thank the reviewer who highlighted that this study was technically well done and that we already described the limits of this work in the discussion of the manuscript.

Reviewer 2 Report

In this study, the authors analyzed the performance of the commercial chromogenic medium, CHOMagar™ LIN-R, for screening of linezolid-resistant gram positive isolates.

The research design is clearly described and the results obtained provide intersting information about the potential application of this medium.

Author Response

In this study, the authors analyzed the performance of the commercial chromogenic medium, CHOMagar™ LIN-R, for screening of linezolid-resistant gram positive isolates.

The research design is clearly described and the results obtained provide interesting information about the potential application of this medium.

Answer: We want to thank the reviewer for his comment.

Reviewer 3 Report

A few comment for the authors that may add to the quality of the Manuscript:

  1. Linezolid (LZD) was the first molecule of the oxazolidinone family that was approved for commercial use in 2000 - please add where or the agency that approved it
  2. Please remove the following This section may be divided by subheadings. It should provide a concise and precise description of the experimental results, their interpretation, as well as the experimental conclusions that can be drawn. 
  3. Can reference be removed from table 1? it is somewhat confusing what does it reffer to. Mentioning 'both studies' throught the text is confusing
  4. This part, and all similar belong to the methods section Regarding the previously reported good performances of the CHROMagar™ LIN-R 151
    on pure bacterial isolates [17] that we confirmed in our validation study, as well as its reliability on positive blood cultures, we decided to test this medium prospectively on nasal swabs that are relevant samples for the screening of LZDR staphylococci. From May to June 2020, nasal screening swabs collected for the screening of MRSA in patients hospitalized at Paul Brousse hospital were included. Results will hardly require a reference.
  5. This part should be moved to discussion Altogether, our results indicate that CHROMagar™ LIN-R medium might be a relevant 180
    screening medium for the detection of LZDR gram positives from clinical samples, especially in medical structures where the prevalence of linezolid resistance is suspected to be high. Of note, such screening media usually possess good sensitivity but might have less specificity. Accordingly, bacterial colonies culturing on such media have to be identified and further subjected to complementary tests to move out false positive results. This is also the case with the CHROMagar™ LIN-R medium for which rapid identification would help to immediately detect the 6.4% of the samples that cultured with irrelevant intrinsically LZDR gram negatives and yeasts. Susceptibility testing
  6. How was the needed sample size determined for validation all other purposes?

Author Response

few comment for the authors that may add to the quality of the Manuscript:

  1. Linezolid (LZD) was the first molecule of the oxazolidinone family that was approved for commercial use in 2000 - please add where or the agency that approved it

Answer: LZD was approved by the US Food and Drug Administration for commercial use. As requested, it has been added in the revised version of the manuscript.

  1. Please remove the following This section may be divided by subheadings. It should provide a concise and precise description of the experimental results, their interpretation, as well as the experimental conclusions that can be drawn. 

Answer: Indeed, we forgot to delete this sentence from the template. It has been deleted in the revised version of the manuscript.

  1. Can reference be removed from table 1? it is somewhat confusing what does it reffer to. Mentioning 'both studies' throught the text is confusing

Answer: As requested, the reference 18 was removed from the table 1. In addition, “both studies” was replaced by “pure isolates and blood cultures”

  1. This part, and all similar belong to the methods section Regarding the previously reported good performances of the CHROMagar™ LIN-R 151 on pure bacterial isolates [17] that we confirmed in our validation study, as well as its reliability on positive blood cultures, we decided to test this medium prospectively on nasal swabs that are relevant samples for the screening of LZDR staphylococci. From May to June 2020, nasal screening swabs collected for the screening of MRSA in patients hospitalized at Paul Brousse hospital were included. Results will hardly require a reference.

Answer: Indeed, this section has been deleted because it was redundant with the Materials and Methods section. Reference 18 is adequate for this study.

  1. This part should be moved to discussion Altogether, our results indicate that CHROMagar™ LIN-R medium might be a relevant 180
    screening medium for the detection of LZDR gram positives from clinical samples, especially in medical structures where the prevalence of linezolid resistance is suspected to be high. Of note, such screening media usually possess good sensitivity but might have less specificity. Accordingly, bacterial colonies culturing on such media have to be identified and further subjected to complementary tests to move out false positive results. This is also the case with the CHROMagar™ LIN-R medium for which rapid identification would help to immediately detect the 6.4% of the samples that cultured with irrelevant intrinsically LZDR gram negatives and yeasts.

Answer: This modification has been performed as requested.

  1. How was the needed sample size determined for validation all other purposes?

Answer: The sample size was determined based on the estimated prevalence of linezolid-resistant isolates that we expected to collect prospectively.

Reviewer 4 Report

Dear Authors,

your work is quite interesting as it is focused on one of the last resort antibiotics. It is characterized by originality and novelty as the medium CHOMagar™ LIN-R was not evaluated on clinical samples. It is also charectized by a high quality of presentation. The results are of high importance and can be easily implemented in the routine flow.

I suggest you to go on some minor changes in order to be published their work.

Specific Comments

L27: note, slight growth (please change the size of the letters)

L41:  as well as

L57:

L80-83: CHROMagar™ LIN-R (CHROMagar, Paris, France) is a selective commercially available medium designed, focused on Linezolid-Resistant Bacteria. It is suitable to identify LZDR gram positives from clinical samples such as nasal swabs used for MRSA screening and rectal swabs performed for the screening of glycopeptide resistant enterococci.

L96-98:

L102: as large and numerous

L103: Please show the comparison between CHROMagar™ LIN-R and Mueller Hinton Agar in Supplementary Figure 1 e.g. add and Table1.

L103-104: This result was independent of the cfr gene content done in a previous work.

L205: “to include” to be included

L212: “Despite with a quite low specificity”. Despite the rather low specificity…

Author Response

Dear Authors,

your work is quite interesting as it is focused on one of the last resort antibiotics. It is characterized by originality and novelty as the medium CHOMagar™ LIN-R was not evaluated on clinical samples. It is also charectized by a high quality of presentation. The results are of high importance and can be easily implemented in the routine flow.

Answer: We want to thank the reviewer for his comment.

I suggest you to go on some minor changes in order to be published their work.

Specific Comments

L27: note, slight growth (please change the size of the letters)

Answer: Modified accordingly.

L41:  as well as L57: ??

L80-83: CHROMagar™ LIN-R (CHROMagar, Paris, France) is a selective commercially available medium designed, focused on Linezolid-Resistant Bacteria. It is suitable to identify LZDR gram positives from clinical samples such as nasal swabs used for MRSA screening and rectal swabs performed for the screening of glycopeptide resistant enterococci.

Answer: Modified accordingly.

L96-98:

Answer: Indeed, we forgot to delete this sentence from the template. It has been deleted in the revised version of the manuscript.

L102: as large and numerous

Answer: Modified accordingly.

L103: Please show the comparison between CHROMagar™ LIN-R and Mueller Hinton Agar in Supplementary Figure 1 e.g. add and Table1.

Answer: Unfortunately, we did not take a photo of Mueller-Hinton plates. Accordingly, we cannot added them to Supplementary Figure 1.

As requested “and Table 1” was added in the revised version of the manuscript.

L103-104: This result was independent of the cfr gene content done in a previous work.

Answer: Modified accordingly.

L205: “to include” to be included

Answer: Modified accordingly.

L212: “Despite with a quite low specificity”. Despite the rather low specificity…

Answer: Modified accordingly.